



# Long-term impacts of mixotrophy on ocean carbon storage: insights from a 10,000-year global model simulation

Marco Puglia[1,2], Thomas S. Bibby[2], Jamie D. Wilson[3], and Ben A. Ward[2]

[1]Istituto Nazionale di Oceanografia e di Geofisica Sperimentale (OGS), Trieste, Italia
[2]School of Ocean and Earth Science, University of Southampton, Southampton, UK
[3]Earth, Ocean and Ecological Sciences, University of Liverpool, Liverpool, UK

**Correspondence:** Marco Puglia (mpuglia@ogs.it)

**Abstract.** Mixotrophs — organisms that combine the use of light and inorganic resources with the ingestion of prey — have been shown in simulations to increase mean organism size and carbon export. These simulations have, however, been limited to decade-long timescales that are insufficient to investigate the impacts of mixotrophy on the ocean's long-term capacity for carbon storage. Here we explore these long-term impacts using a low-resolution ocean model that resolves important feedbacks

between surface ecology and the ocean interior over multi-millennial periods. The model was compared in two configurations: one with a strict distinction between phytoplankton and zooplankton populations and one in which all populations were assumed to be capable of mixotrophy. Consistent with earlier studies, we found that increased carbon and nutrient export associated with mixotrophy was rapidly established within the first few years of the simulation and robust over long time scales. However, we also found that these increases were partially offset over longer time scales by a decline in dissolved inorganic

carbon and nutrients entering the deep ocean via the sinking of surface waters. Over the 10,000 year duration of the simulations, we found that ecologically-driven changes in C export increased the oceanic C inventory by up to 626 Pg, and that this was partially offset by decline of 149 Pg in the preformed C inventory, leaving a net increase of up to 477 Pg C (1.3 %).

## 1   Introduction

Plankton communities lie at the foundation of marine food webs, indirectly supporting human populations through fisheries

(Hollowed et al., 2013) and playing a central role in global biogeochemical cycles (Falkowski et al., 1998; Worden et al., 2015). Despite the frequent conceptual and practical division of plankton into autotrophic phytoplankton and heterotrophic zooplankton, especially in past modelling studies (Azam et al., 1983; Fasham et al., 1990; Burkholder et al., 2008), these modes of nutrition are not mutually exclusive and an increasing number of species are recognized as being mixotrophic (Stoecker, 1998; Hartmann et al., 2012; Stoecker et al., 2017).

Modelling studies have shown that mixotrophs can impact marine ecosystem structure and function (Stoecker, 1998), increasing the efficiency with which nutrients and energy are passed through the food web (Mitra et al., 2014; Ward and Follows, 2016). This may lead to increased primary production (Baretta-Bekker et al., 1998; Hammer and Pitchford, 2005; Mitra et al., 2014), larger organisms and increased export (Ward and Follows, 2016; Stoecker et al., 2017). At the global scale, Ward and Follows (2016) showed that mixotrophy allowed for increased carbon export from the ocean surface, compared to an equivalent





simulation without mixotrophy. In their simulations photosynthesis supported by the mixotrophic ingestion of prey provided an additional source of carbon for the same supply of limiting nutrients. This increase sustained the growth of larger and faster sinking organisms that were relatively enriched in carbon, leading to increased export of carbon to depth.

The global modelling study of Ward and Follows (2016) was restricted to a time frame between 10 and 15 years. This is sufficient time for the ecological community to stabilize given the modelled nutrient supply from depth, but it was not sufficient
to resolve any feedbacks that might occur as the ecological community alters the deep ocean nutrient and carbon inventories over thousands of years (DeVries et al., 2012). For example, the presence of larger and thus faster-sinking plankton (Sournia, 1982) could potentially lead to a deeper sequestration of nutrients and an associated decline in primary production at the surface.

To investigate this issue, this article investigates the impacts of mixotrophy in a global model of the marine plankton com-
munity, using the coarse-resolution EcoGEnIE model (Ward et al., 2018). In the following, we describe the global ocean model and the experimental design. We then present results that illustrate the response of the ocean ecosystem at the surface and at depth over a range of time scales from 1 to 10,000 years. We show that the increased organism size and carbon-to-nutrient ratios attributable to mixotrophy are robust to any long-term feedbacks with the ocean interior. However, while the expected ecological shifts led to a net increase of carbon sequestration at large scales, mixotrophy also led to a decline in carbon seques-
tration in the North Atlantic. We discuss the mechanisms underpinning this response and explore the sensitivity of our findings to the strength of unknown physiological costs associated with mixotrophy.

## 2   Global Ecosystem Model

We use the EcoGEnIE (Ward et al., 2018) model, which is an ecological extension of cGEnIE ("carbon-centric Grid Enabled Integrated Earth system model"; Ridgwell et al., 2007). EcoGEnIE is an Earth System Model of Intermediate Complexity
(Claussen et al., 2002) with a low spatial resolution that facilitates long simulations (on the order of 10,000 years) to be achieved within hours or days (depending on the ecological complexity of the model) on a personal computer or workstation.

The ecosystem model resolves the plankton community into a number of different size classes and each size class may be resolved into a number of functional groups (e.g. phytoplankton, mixotrophs, zooplankton). Each functional group has an associated set of size-dependent traits. For example, phytoplankton take up inorganic nutrients and use light to photosynthesise
(Geider et al., 1998), zooplankton obtain resources by consuming smaller prey, while mixotrophs can combine these traits to varying degrees. Several traits are size dependent (Edwards et al., 2012; Marañón et al., 2013; Ward et al., 2017). For example, biomass-specific nutrient affinities and uptake rates are highest in the smallest phototrophic plankton, while biomass-specific maximum photosynthetic rates decline either side of a peak at approximately 6 μm diameter (Marañón et al., 2013). Zooplankton graze most efficiently on prey that are 10 times smaller than themselves in length, with maximum grazing rates
increasing with decreasing organism size (Hansen et al., 1997). A full description of EcoGEnIE can be found in Ward et al. (2018).





## 2.1 Ecosystem Configurations

To assess the impact of mixotrophy on the global ocean ecosystem, we compared five different configurations of EcoGEnIE. All configurations resolved eight plankton size classes of 0.6, 1.9, 6, 19, 60, 190, 600 and 1900 μm equivalent spherical diameter (ESD). In the first "Two-Guild" simulation, each size class was divided into two populations: one strictly autotrophic phytoplankton and one strictly heterotrophic zooplankton. This simulation was the standard configuration used in Ward et al. (2018) and is used here as a non-mixotrophic control, against which the effects of mixotrophy in other simulations can be assessed.

A second "Mixotrophic" configuration did not divide the eight size classes into individual functional groups. Instead, a single mixotrophic population in each size class was allowed to take up nutrients, photosynthesise and consume prey. The populations in the Mixotrophic configuration do not incur any penalty for the ability to combine autotrophic and heterotrophic traits and are simultaneously assigned the same traits as both the phytoplankton and zooplankton populations in the Two-Guild configuration.

The assumption that there is no physiological cost associated with mixotrophy is not necessarily realistic (Edwards et al., 2023a), but it does allow estimation of the maximum possible impacts of mixotrophy in the modelled ecosystem (Ward and Follows, 2016). To test the sensitivity of the model to potentially more realistic (albeit poorly constrained) costs associated with mixotrophy, we also tested three additional "Trade-Off" configurations. These configurations included three trophic strategies, with one phytoplankton, one mixotroph and one zooplankton in each size class. The maximum nutrient uptake rates, maximum photosynthetic rates and maximum grazing rates of the mixotrophic populations were then scaled by a fixed trade-off factor. In these three simulations, maximum rates of nutrient uptake, photosynthesis and grazing were downgraded to 60, 50 and 40 % of the specialist values used in the Two-Guild configuration. All five model configurations are summarized in Table 1.

**Table 1.** *Configuration of the five ecological model configurations with the degree to which maximum rates of nutrient uptake, photosynthesis and grazing where decreased in the mixotrophic populations.*

| Configuration | Trade-Off factor | Number of populations | | |
| --- | --- | --- | --- | --- |
| | | Phytoplankton | Mixotrophs | Zooplankton |
| Two-Guild | - | 8 | - | 8 |
| Mixotrophic | 100% | - | 8 | - |
| Trade-Off 60 | 60% | 8 | 8 | 8 |
| Trade-Off 50 | 50% | 8 | 8 | 8 |
| Trade-Off 40 | 40% | 8 | 8 | 8 |

*Note.* The trade-off factor scales the maximum rates of nutrient uptake, photosynthesis and grazing in the mixotrophic populations. Phytoplankton have nutrient uptake and photosynthetic rates set to 100 % of the default values, while the maximum grazing rate is set to zero. The opposite is true for the zooplankton populations (Ward et al., 2018).



## 2.2 Simulations

To assess the long-term biogeochemical effects of changing the ecosystem configuration through time, the model was first spun-up for 10,000 years with the Two-Guild configuration. All model configurations (including the Two-Guild configuration) were then run for a further 10,000 years from this spin-up, so that changes relative to the Two-Guild configuration can be tracked as they develop through time. 10,000 years was sufficient time for the ocean to reach a repeating annual cycle with no changes in the global annual averages for the model state variables. In all cases the atmospheric $CO_2$ concentration was fixed at 278 ppm, as in Ward et al. (2018).

We initially compare the Mixotrophic simulation to the Two-Guild (control) simulation to identify an upper limit for the possible impacts of mixotrophy on the modelled system. Further simulations are then used to explore the sensitivity of these effects to the assumed strength of any physiological costs associated with mixotrophy.

## 3 Results

Prior global modelling on a decadal timescale (Ward and Follows, 2016) has suggested that increased trophic transfer efficiency associated with mixotrophy can increase mean plankton size and global carbon export. Here we test whether these effects are also seen in the EcoGEnIE model and examine whether the simulated changes are robust to biogeochemical feedbacks with the ocean interior occurring over millennial timescales.

### 3.1 Global-scale effects

Figure 1 shows the global mean carbon biomass within each model size class throughout the Two-Guild and Mixotrophic configurations. With different years of the simulations shown with different colours, it is clear that distinctive size distributions are established within the first year of each simulation, and that these distributions remain relatively stable throughout the 10,000 years of each simulation. At the end of the two simulations, the global biomass-weighted geometric mean size is 22.44 $\mu$m ($\overset{\times}{\div}6.21$) in the Two-Guild configuration, increasing to 54.85 $\mu$m ($\overset{\times}{\div}7.42$) in the Mixotrophic configuration.

The influence of mixotrophy on the surface ecosystem is further explored in Fig. 2, which shows the change in six key ecosystem characteristics from the Two-Guild to the Mixotrophic configuration at four time points. In terms of ESD, the switch to mixotrophy drives a universal increase in biomass-weighted geometric mean size (Fig. 2a–d), with the largest changes in the most productive regions. The effects of mixotrophy on the particulate carbon and phosphorous export fluxes are more complicated, with a differential response at high and low latitudes (Fig. 2e–l). Export of particulate organic carbon generally increases between 60° north and south, with a decline in the Southern Ocean. Export of particulate organic phosphorus shows almost the opposite trend, increasing in the Southern Ocean, the North Atlantic and the Northwest Pacific, while remaining relatively unaffected at lower latitudes.

While increased plankton size is associated with deeper sinking, the contrasting responses of C and P export suggest that particle size is not the dominant mechanism by which mixotrophy affects the magnitude of export in the model. Changes in





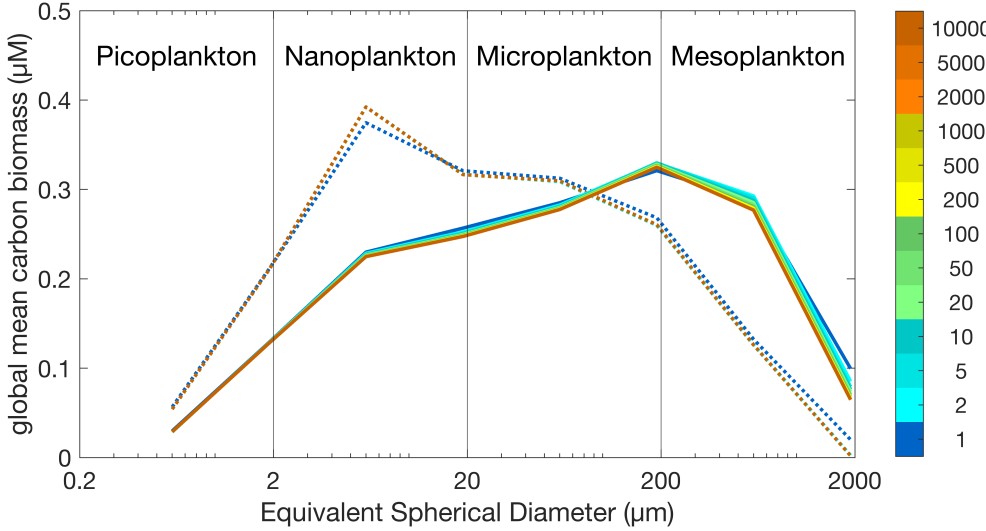

**Figure 1.** Global plankton size distributions in the Two-Guild (dotted lines) and Mixotrophic (solid lines) configurations. Colours correspond to the simulation year, as indicated by the colour scale.

particulate export flux related to mixotrophy appear to be more strongly associated with changes in plankton uptake stoichiometry. If we define the community C:P uptake ratio as the community-integrated ratio of inorganic carbon fixation to inorganic phosphate uptake, Fig. 2(m–p) suggests that a higher C:P uptake ratio among mixotrophic plankton at low latitudes allows them to export more carbon there, while a lower C:P uptake ratio at high latitudes hinders carbon export. Conversely, the inverse (P:C) uptake ratio seems to play a similar role in regulating P export.

An exception to this first order pattern occurs in some regions, such as the North Atlantic and the sub-Antarctic zone of the Southern Ocean, where C export increases despite a decline in the C:P uptake ratio. These regions are also associated with very large increases in plankton mean size, which appears to be driving an increased C export flux despite a decreased C:P uptake ratio.

While differences in surface $PO_4$ develop early in the simulation and remain relatively stable throughout (Fig. 2u–x), surface DIC takes somewhat longer for the full differences to develop (Fig. 2q–t). While $PO_4$ is a lot lower in the mixotrophic simulation at high latitudes, DIC initially declines at low latitudes (suggestive of an ecological effect), while lower concentrations develop at higher latitudes over hundreds to thousands of years (suggestive of a feedback with the ocean circulation).

Regardless of these spatial trends, it is clear that the changes to ESD, export and C:P uptake ratios established inside the first ten years remain relatively stable throughout the remainder of the simulation (relative to the spatial variability within each time step). The only exceptions to this are P export in the Atlantic Ocean and South East Pacific, which appears to decline slightly over longer timescales, and surface DIC, which declines over a much larger region on long time scales. In general, this indicates that the previously reported ecological and biogeochemical effects of mixotrophy at the surface (Ward and Follows,





**Figure 2.** Changes between the Mixotrophic and Two-Guild configurations at different time points. Values greater than one (red) indicate regions where mixotrophy increased the variable in question, whereas values less than one (blue) indicate a decline. First row (a-d): equivalent spherical diameter of all plankton. Second row (e-h): particulate organic carbon export. Third row (i-l): particulate organic phosphorus export. Fourth row (m-p): community autotrophic C:P uptake ratio. Fifth row (q-t): dissolved inorganic carbon. Sixth row (u-x): dissolved $PO_4$. All rows show Mixotrophy-Two-Guild, except C:P uptake, which is Mixotrophy÷Two-Guild.





2016) are largely robust to feedbacks from the ocean interior, although there may be some internal processes that are worthy
of investigation.

## 3.2 Impacts on the ocean interior

Figure 3 shows how the mean vertical distribution of DIC and $PO_4$ develop through time in each of the Pacific, Atlantic and In-
130 dian Ocean basins. As both model configurations were initialised from an equilibrated spin-up of the Two-Guild configuration,
these vertical distributions do not change in the Two-Guild simulation. Contrary to Fig. 1, in which large and stable differ-
ences between the Two-Guild and Mixotrophic are established within the first year of the simulations, noticeable difference
between the two configurations only begin to emerge on multi-decadal timescales (green colours). These continue to develop
over multi-centennial timescales (yellow colours), and have largely stabilised by multi-millennial timescales (orange colours).

Within the Pacific and Indian Oceans, mixotrophy has the straightforward effect of increasing DIC storage at depth over
long time scales, with a slight decrease in DIC at the surface. The response within the Atlantic Ocean is more complex. Over
the first 500 years, mixotrophy leads to increased DIC storage at all depths below the thermocline, but after about 1000 years,
there is a decline in DIC storage between about 1,500 and 3,500 m.

The vertical $PO_4$ distributions show a similar, but more subtle response to mixotrophy. $PO_4$ concentrations increase slightly
at depth in the Pacific and Indian Oceans, but decline slightly in the Atlantic.

The differential response in the Atlantic Ocean occurs at depths associated with the North Atlantic Deep Water (NADW)
(Toggweiler and Key, 2001), which suggests some feedback with the ocean circulation. This is explored in Figures 4 and 5,
which show the temporal evolution of differences in meridional averaged chemical profiles for DIC and $PO_4$.

The patterns shown in Fig. 4 reflect what is shown by the vertical profiles in Fig. 3. Early in the simulation the increased
export of POC at low latitudes (Fig. 2i) leads to a corresponding downward shift in DIC in all three ocean basins, with less
DIC stored near the surface and a corresponding accumulation of DIC at depth. This is consistent with the deeper sinking of
the larger and more carbon-enriched organic matter produced by the mixotrophic community (Fig. 2).

This downward shift continues to develop through time in the Indian and the Pacific oceans, with mixotrophy causing a
widespread increase in DIC storage in the ocean interior. The Atlantic ocean, by contrast, shows a more complex response
across the duration of the simulation. While the South Atlantic shows a similar increase in C storage at depth, the North
Atlantic shows a decline in DIC storage associated with ventilation of the ocean interior by the NADW.

Corresponding changes in the oceanic $PO_4$ distribution are shown in Fig. 5. As was the case for DIC, the initial response
within the first 10 to 100 years of the simulations is similar across all three ocean basins, with a deepening of the $PO_4$ profile
at high latitudes. This response continues to develop within the Pacific and Indian Oceans, while the Atlantic Ocean deviates
over longer timescales, with less $PO_4$ at stored depth in the Mixotrophic Simulation. This decline is much more widespread
than was seen for DIC (Fig. 4h).



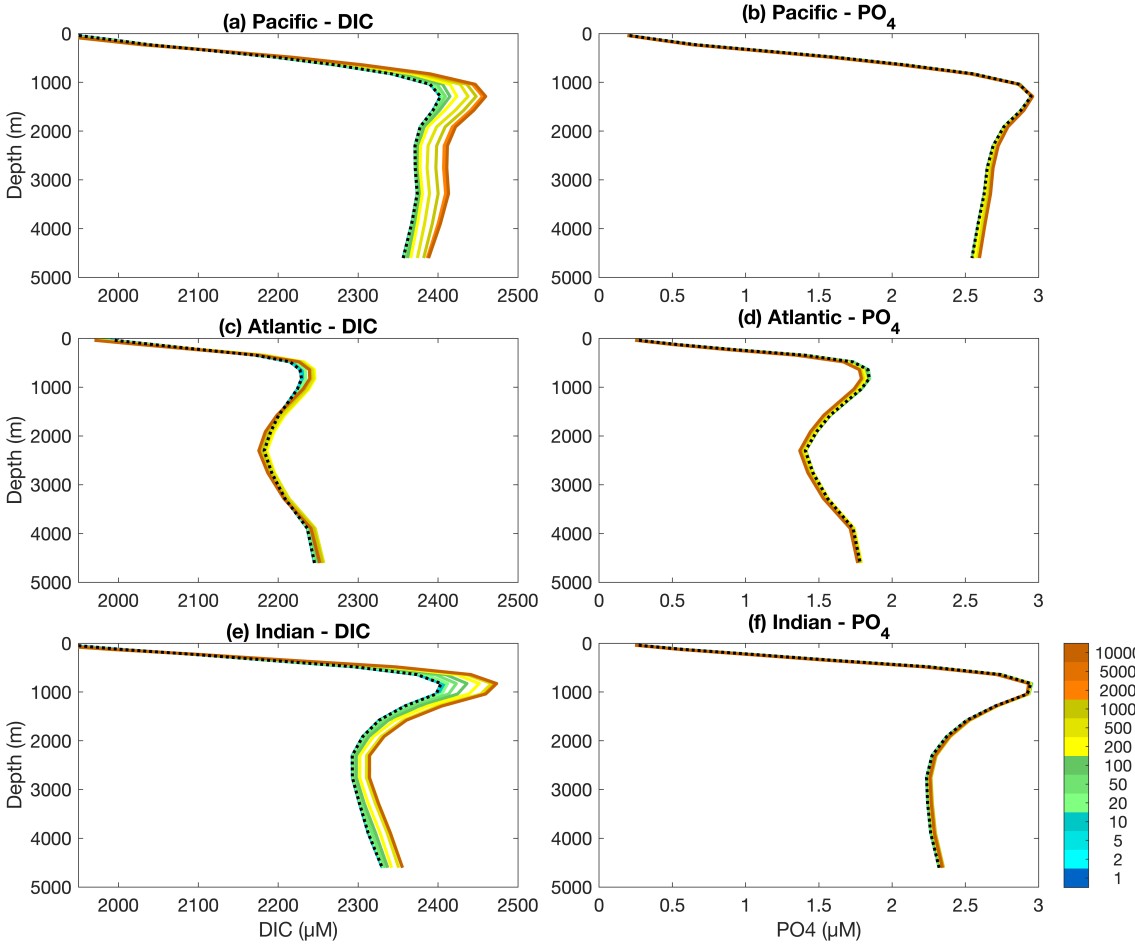

**Figure 3.** Basin-averaged depth profiles of dissolved inorganic carbon (DIC) and phosphorous (PO$_4$) in the three main ocean basins for the the Two-Guild (black dotted line) and Mixotrophic (coloured solid lines) simulations. Simulation years for the Mixotrophic simulation are indicated by colours, as in Fig. 1. Two-guild simulation is plotted in black because the internal profiles do not change from the initial state.



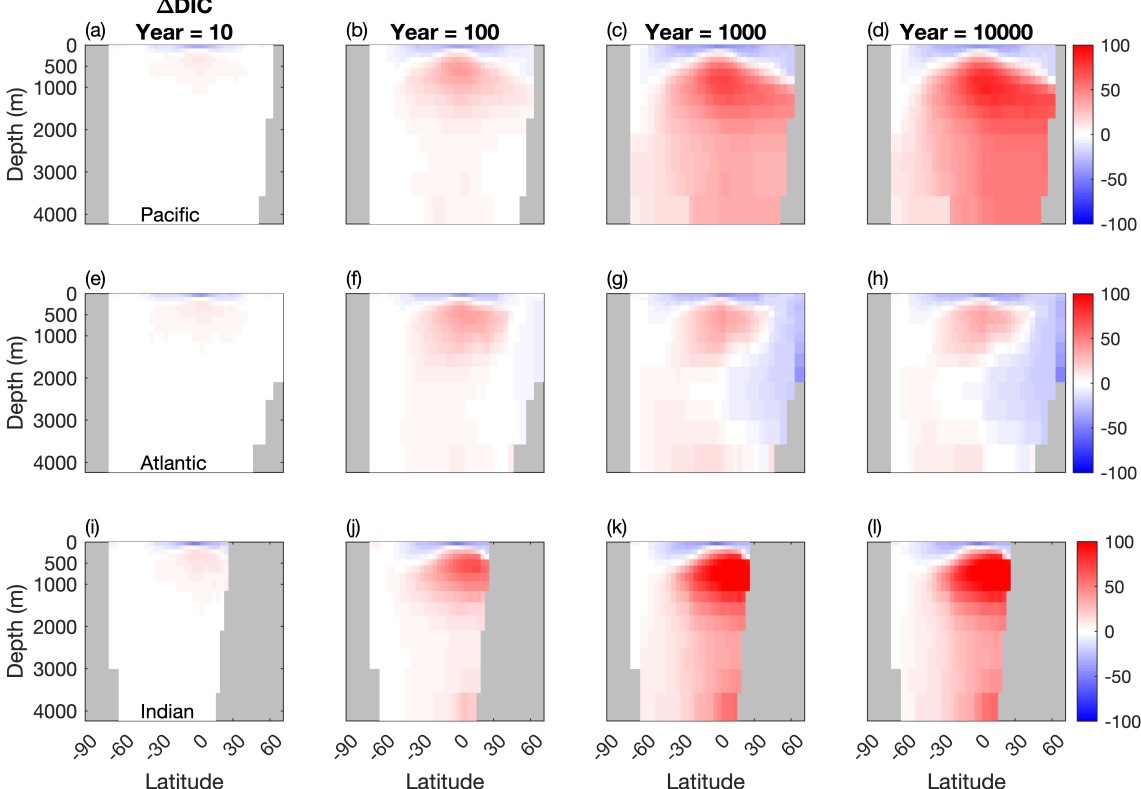

**Figure 4.** Temporal evolution of differences in the meridional averaged DIC concentration between the two configurations ($\mu$M). **(a, b, c)** Atlantic basin, **(d, e, f)**, Indian basin, and **(g, h, i)** Pacific basin. Blue: less DIC stored in the Mixotrophic configurations. Red: more DIC stored in the Mixotrophic configuration.

### Remineralised and preformed nutrients

The distribution of nutrients within the ocean interior is primarily influenced by two main processes, namely the local remineralisation of organic matter and the transport of 'preformed' inorganic nutrients from the ocean surface (Ito et al., 2005). The effect of mixotrophy on these two processes is shown for DIC and $PO_4$ in Figures 6 and 7.

Figure 6 shows that while mixotrophy drives increased internal DIC storage through export and remineralisation, it has an opposite effect on the contribution from preformed nutrients (Table 2). Increased uptake and export of carbon at lower latitudes (Fig. 2h) drives a decline in surface DIC in these regions (Fig. 2t), and thus less DIC is carried into the ocean interior as those surface waters are subducted into the ocean interior. With relatively little deep water formation in the Pacific and Indian Oceans, the declining contribution from preformed nutrients has little impact on the interior DIC distribution, but formation and subduction of NADW in the North Atlantic provides a pathway into the ocean interior.

A similar effect is seen for $PO_4$, although in this case the impact on preformed $PO_4$ is felt most strongly at higher latitudes, where increased uptake and export of phosphorus (Fig. 2l) drive a decline in surface (and thus preformed) $PO_4$ (Fig. 2x).




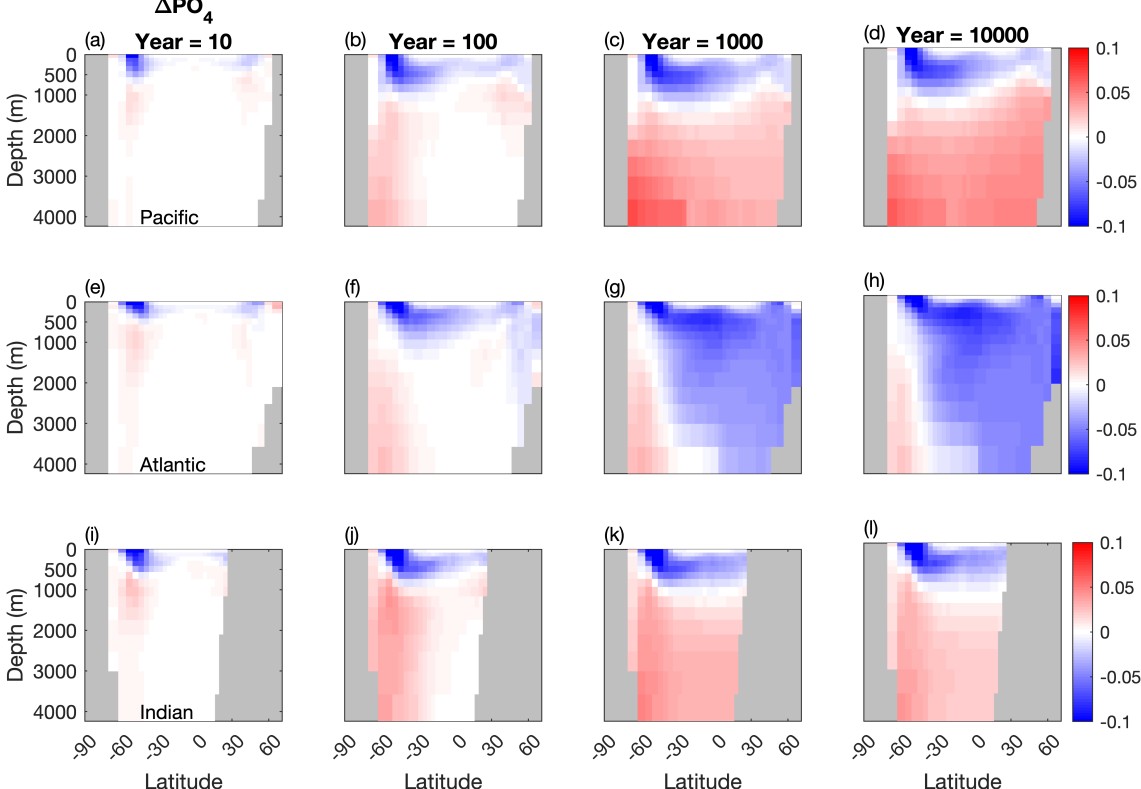

**Figure 5.** Temporal evolution of differences in the meridional averaged $PO_4$ concentration between the two configurations ($\mu$M). **(a, b, c)** Atlantic basin, **(d, e, f)**, Indian basin, and **(g, h, i)** Pacific basin. Blue: less $PO_4$ stored in the Mixotrophic configurations. Red: more $PO_4$ stored in the Mixotrophic configuration.

The impact of mixotrophy also has a much stronger relative effect on preformed $PO_4$ than on preformed DIC. This is likely because the increased drawdown of DIC is tempered to some extent by increased dissolution of $CO_2$ from the model's atmosphere, which has a fixed $CO_2$ concentration of 278 ppm. This does not occur for $PO_4$, for which the model ocean has a fixed inventory, with no atmospheric exchange to compensate for increased drawdown.

## 4 Discussion

On timescales greater than approximately one year, the amount of exogenous nutrients supplied to the ocean surface is often assumed to be balanced by the vertical export of biomass (Hain et al., 2014). In principle, under this assumption, carbon export is set by the physical supply of 'new' nutrients from below. In practice, however, the amount of carbon sequestered may be modified by both the size and stoichiometry of plankton. Larger plankton produce larger detrital particles (Small et al., 1979) that account for the majority of the vertical flux in the water column (McCave, 1975), while more carbon-rich organic matter exports more carbon per mole of limiting nutrient supplied.





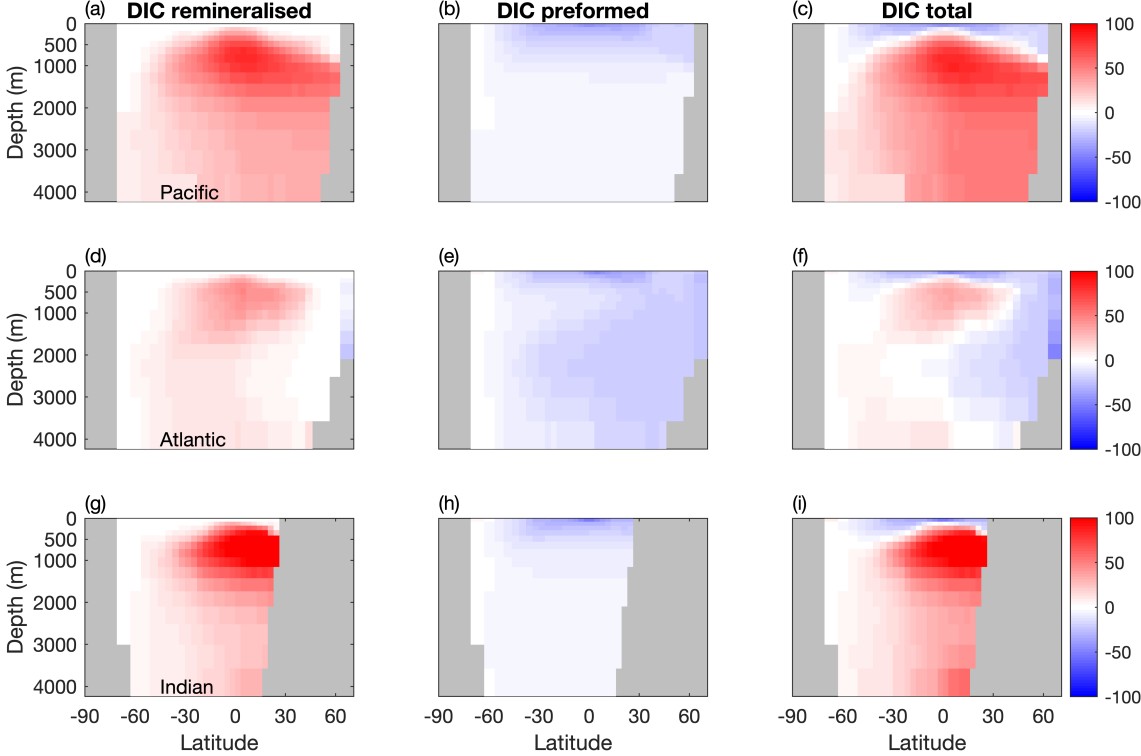

**Figure 6.** Differences in remineralised, preformed and total DIC between the two configurations ($\mu$M) in the Pacific, Atlantic and Indian Basins. Blue: less DIC stored in the Mixotrophic configurations. Red: more DIC stored in the Mixotrophic configuration.

**Table 2.** Comparison of Preformed, Exported, and Total DIC (Pg) across the different simulations. The $\Delta$ columns are the differences between the four mixotrophic simulations and the Two-Guild simulation.

|  | Exported | $\Delta$ Exported | Preformed | $\Delta$ Preformed | Total | $\Delta$ Total |
|---|---|---|---|---|---|---|
| Two-Guild | 2449 | - | 34201 | - | 36650 | - |
| Mixotrophic | 3076 | 626.7 | 34052 | -149.8 | 37127 | 476.9 |
| Trade-Off 60 | 2667 | 217.8 | 34131 | -70.0 | 36798 | 147.8 |
| Trade-Off 50 | 2612 | 162.8 | 34135 | -66.0 | 36747 | 96.8 |
| Trade-Off 40 | 2557 | 107.8 | 34144 | -57.6 | 36700 | 50.2 |

Mixotrophy has been shown influence carbon export through its effects on plankton size and stoichiometry. Using a global ocean circulation model with one-degree resolution, Ward and Follows (2016) showed that mixotrophy allows photosynthesis to occur above the basal trophic level, supported by additional nutrients acquired from prey. At low latitudes in particular (where nutrients are more scarce), this increased the ratio of photosynthesis to inorganic nutrient uptake, and allowed more biomass to be passed up the food web to larger and faster sinking organisms.



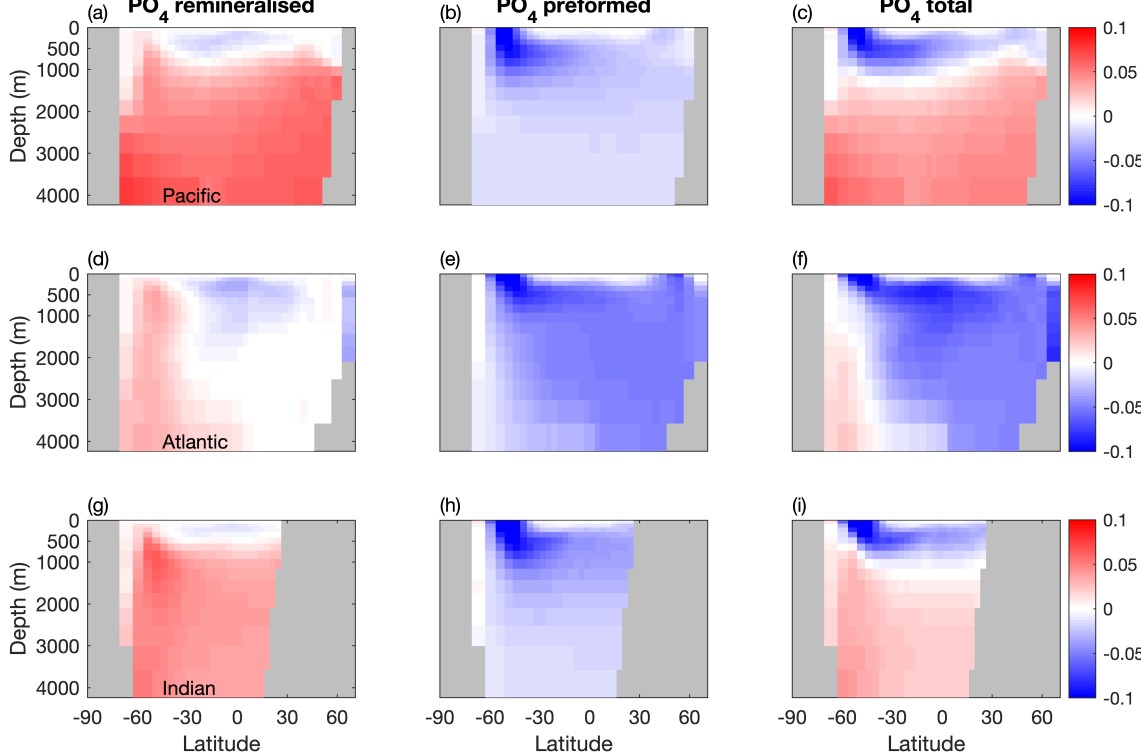

**Figure 7.** Temporal evolution of differences in the remineralised and preformed PO$_4$ between the two configurations ($\mu$M). **(a, b, c)** Atlantic basin, **(d, e, f)**, Indian basin, and **(g, h, i)** Pacific basin. Blue: less PO$_4$ stored in the Mixotrophic configurations. Red: more PO$_4$ stored in the Mixotrophic configuration.

While the enhanced sinking of larger and more carbon-enriched material was associated with increased carbon export, the 10-15 year timescale of the simulations used by Ward and Follows (2016) did not allow investigation of how these changes might impact ocean carbon sequestration. An unanswered question was what would happen to community structure and function once long-term feedbacks with the ocean circulation have been accounted for. Specifically, could long-term changes in oceanic DIC and nutrient distributions feed back and modify the initial ecological response. In this article the lower resolution EcoGEnIE
model was used to simulate the system over the timescales required for these feedbacks to be resolved.

The results presented above confirm that the change from the Two-Guild to the Mixotrophic configuration rapidly affects ecosystem structure and function, and further suggest that these changes remain stable even after long-term feedbacks between the surface and ocean interior have been accounted for.

All plankton must acquire energy and a number of bioavailable elements (C, N, P, Fe, etc) to grow and reproduce. For
specialist phytoplankton, these resource come from photosynthesis and the uptake of inorganic nutrients. When light or any one inorganic nutrient becomes sufficiently scarce, the uptake of all elements is restricted. Mixotrophs, by contrast, can overcome the scarcity of individual inorganic resources through predation. This has previously been discussed as a mechanism by which





mixotrophy allows more photosynthesis for the same supply of limiting nutrients, allowing more carbon to be exported to depth (Ward and Follows, 2016).

In the results presented above, it appears that the same mechanism allows greater uptake and export of $PO_4$ at high latitudes. Whereas the scarcity of light and dissolved iron is enough to limit the uptake of non-limiting $PO_4$ by phytoplankton in these regions, carbon and iron acquired from prey allow more inorganic $PO_4$ uptake in the Mixotrophic model.

The direct consequence of these changes is that the Mixotrophic model exports more carbon at low latitudes and more phosphorus at high latitudes than the Two-Guild model. These effects emerge on very short yearly time scales and remain
robust over millennia.

These results are largely consistent with the findings of Ward and Follows (2016), but here we also see the emergence of an indirect effect over centuries to millennia. Specifically, the additional DIC and $PO_4$ removed by mixotrophs at the surface can lead to a decline in the deep ocean carbon and phosphate inventories in regions where surface waters are transported to depth. This is apparent in the influence of mixotrophs on preformed DIC and phosphate shown in Figures 6 and 7.

The subduction of carbon and phosphorus depleted surface waters into the ocean interior partially counteracts the direct effects of mixotrophy on the biological carbon pump (Table 2), resulting in lower DIC and $PO_4$ storage in the North Atlantic.

In general however, mixotrophy drives DIC and $PO_4$ deeper into the water column. While the former represents increased oceanic carbon sequestration, the latter has the potential to impede oceanic production, if less $PO_4$ is being supplied the ocean surface. By running EcoGEnIE for 10,000 years, we were able to test whether such a feedback could temper the initial
ecosystem response seen on annual to decadal timescales. In general, the consistent responses of the surface ecosystem through time suggest that the importance of any such feedbacks is limited, with a few exceptions, as noted above.

**Sensitivity to trade-offs**

The results presented and discussed above make the unrealistic assumption that mixotrophs incur no physiological costs for combining autotrophic and heterotrophic traits (Edwards et al., 2023a, b). It is therefore important to assess the potential impact
of any physiological costs associated with mixotrophy.

Figure 8 shows how different potential costs of mixotrophy can moderate the ecological and biogeochemical impacts discussed above. As might be expected, the overall impact of mixotrophy on organism size and carbon export decreased when mixotrophs were subject to stronger physiological costs. While mixotrophy causes organism size and export to change in different directions in different regions, in general, the simulations revealed that a weaker trade-off (i.e. a lesser penalty associated
with mixotrophy) made positive differences more positive (e.g. export production at low latitudes and organism size increased by more) and made negative differences more negative (e.g. export production at high latitudes declined by more). This pattern suggests that the results presented above constitute an upper limit to the impacts of mixotrophy and that the influence of mixotrophy on the carbon cycle depends on the strength of any associated trade-offs.



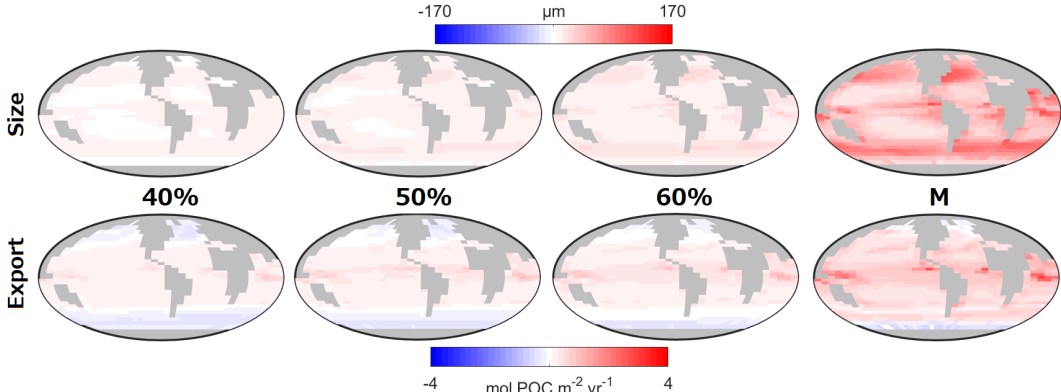

**Figure 8.** Impact of the trophic trade-off on size and carbon export. The maps are the difference between the Mixotrophic/Trade-Off configurations and the Two-Guild configuration. The percentage refers to the Trade-Off factor (Table 1). "M" refers to the Mixotrophic configuration and these last two panels are identical to Fig. 2(d, h), but for the colour bar used for size.

## 5   Conclusions

The long-term simulations presented here confirm that mixotrophy drives increased mean organism size and carbon export, even after long-term feedbacks between surface ecology and the biogeochemical inventory of the ocean interior have been accounted for. While there are some caveats associated with a long-term feedback associated with the transport of preformed DIC and $PO_4$ through the NADW, this does not seem to qualitatively affect the capacity for mixotrophs to increase oceanic carbon storage as a consequence of their increased trophic transfer efficiency.

*Author contributions.*  MP and BW conceptualized the study and performed the formal analysis. MP curated the data. BW and JW developed the methodology. MP, BW, and TB wrote the first draft of the manuscript. MP, BW, TB, and JW contributed to the review and editing of the manuscript. All authors contributed to the interpretation of the results and approved the final version of the manuscript.

*Competing interests.*  The authors declare that they have no conflict of interest.

*Acknowledgements.*  The authors would like to thank CM Moore (University of Southampton) for providing helpful feedback on earlier
versions of the manuscript.

*Financial support.*  BW and MP gratefully acknowledge support from the Royal Society.



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
