# Peer review of "Long-term impacts of mixotrophy on ocean carbon storage: insights from a 10,000-year global model simulation"

_EGUsphere, 2025_

## Referee Comment (RC1)

**Review comments for the manuscript "Long-term impacts of mixotrophy on ocean carbon storage: insights from a 10,000-year global model simulation" by Puglia et al.**

This article investigates the impact of mixotrophy on the ocean's long-term DIC storage. They do so by running a mixotrophy plankton model in a low-resolution ocean model for 10000 years and comparing the outcome with a run with strict phyto- and zooplankton (without mixotrophs). The results show that in the mixotrophic case, there was an increased carbon and nutrient export. This increase in export resulted on an overall increase in remineralized DIC and PO4 in the ocean interior, but a decrease in preformed nutrients, which became apparent in the north Atlantic. All in all, mixotrophy resulted in a net increase in the ocean carbon storage.

The article is interesting as it goes one step further to the carbon export metric and looks at the ecological effects on the overall ocean DIC storage, which is often omitted. Results are clear and address the stated objective. My comments center on text clarifications, especially regarding the caption of some figures. I also suggest adding some figures in an appendix to make the results more comprehensive.

Comments:

L32-33 "could potentially lead to a deeper sequestration of nutrients and an associated decline in primary production at the surface." An image showing the changes in NPP and total biomass with time would also be nice (perhaps in an appendix). Just to confirm that indeed there is no change in any of these metrics by the end of the 10K years.

L39-40 "mixotrophy also led to a decline in carbon sequestration in the North Atlantic." Consider mentioning that this decline is due to a decrease in preformed nutrients (to be consistent with the abstract, as now this sounds like a new result, and the reader does not know why this is relevant).

L55 it would be nice to provide a description of how stoichiometry is modeled. Mostly because this seems to be an important driver of the observed results. There is no need to write all the equations of the model, but perhaps some equations showing mixotrophic growth and stoichiometry would be great to improve the understanding without having to read the methods in Ward et al 2018. Otherwise, a paragraph explaining how/why

mixotrophs can have different stoichiometries would be good (and trying to link this to the differences observed between low and high latitudes would be even better).

Figure 2: caption needs a more detailed explanation. I guess ESD is derived from the depth-integrated biomass? At which depth approximately is carbon export estimated? Are DIC and PO4 concentrations for the surface layers only? Or are they integrated over a specific depth range or the entire water column?

Figure 2 caption "Values greater than one (red) indicate regions where mixotrophy increased the variable in question, whereas values less than one (blue) indicate a decline." This seems to only apply to panels m-p, the others are centered on 0. Also, I guess the colorbar in panels m-p is in log scale(?) This makes it a bit tricky to read the magnitude of change other than 10 times smaller/larger, because the ticks are not evenly distributed and are not shown in the colorbar. Consider adding some ticks or more labels (in-between 10, 0 and 0.1).

Figure 2, it would be good to have a figure (perhaps in an appendix) showing the base-line values of each panel (e.g. the actual ESD values across latitudes for the mixotrophic run and for the separate guild run etc), so that the reader knows whether the absolute differences shown in the figure are large or not. Otherwise, showing the relative differences (e.g. in percent change) could work too.

L131 "Contrary to Fig. 1, in which large and stable differences between the Two-Guild and Mixotrophic are established within the first year of the simulations, noticeable difference between the two configurations only begin to emerge on multi-decadal timescales (green colours)." This sentence is somewhat confusing, isn't the difference simply because these two figures show two different things? i.e. figure 1 shows surface only while figure 3 depth-resolved differences? I guess the sentence is just confusing and needs to be rephrased (i.e. instead of saying that the difference is between the figures, mention that the difference is between surface and deep ocean).

L136 "The response within the Atlantic Ocean is more complex. Over the first 500 years, mixotrophy leads to increased DIC storage at all depths below the thermocline, but after about 1000 years, there is a decline in DIC storage between about 1,500 and 3,500 m." but these differences are barely noticeable. Is it really that relevant? Or am I missing something? I guess the interesting point here is that the Atlantic Ocean barely sees any difference while the other two oceans show larger differences.

Figure 6. A bit subtle, but it seems to me that DIC total in panel c is "redder" than DIC remineralised from panel a, is this correct? If DIC preformed decreases, shouldn't DIC total be lower than DIC remin?

Table 2 caption. Mention that these differences correspond to the ones found at the end of the 10K years. Also, for consistency with the other figures, switch "exported" DIC with "remineralised" DIC. A similar table showing the differences in the exported carbon and PO4 at a given depth would also have been interesting.

---

## Author Response (AR1)

**RC1**

*-L32-33 "could potentially lead to a deeper sequestration of nutrients and an associated decline in primary production at the surface." An image showing the changes in NPP and total biomass with time would also be nice (perhaps in an appendix). Just to confirm that indeed there is no change in any of these metrics by the end of the 10K years.*

We have added a new Figure A1 to a new *Appendix* section. This shows the temporal development of globally-averaged surface PO4, DOC and POC export, as well as globally-integrated regenerated, preformed and total DIC.

*-L39-40 "mixotrophy also led to a decline in carbon sequestration in the North Atlantic." Consider mentioning that this decline is due to a decrease in preformed nutrients (to be consistent with the abstract, as now this sounds like a new result, and the reader does not know why this is relevant).*

We have added the following text to line 38: "*However, while the expected ecological shifts led to an increase of biological carbon export at large scales, this increase was partially offset by a decline in the amount of preformed dissolved inorganic carbon entering the ocean interior the North Atlantic via physical pathways.*"

*-L55 it would be nice to provide a description of how stoichiometry is modeled. Mostly because this seems to be an important driver of the observed results. There is no need to write all the equations of the model, but perhaps some equations showing mixotrophic growth and stoichiometry would be great to improve the understanding without having to read the methods in Ward et al 2018. Otherwise, a paragraph explaining how/why mixotrophs can have different stoichiometries would be good (and trying to link this to the differences observed between low and high latitudes would be even better).*

We have added the following text at line 58:

> *Flexible cellular stoichiometry is included in the model as each population is represented in terms of its C, P and Fe biomass. The proportion of these elements may change within the cell as they are acquired through photosynthesis, the uptake of inorganic nutrients, and/or the ingestion of prey. Ward & Follows (2016) showed that mixotrophs can achieve the same net growth rate with lower internal reserves of P and Fe, supported by their additional flexibility to gain carbon from both photosynthesis and grazing. Mixotrophic ecosystems are therefore able to sustain higher carbon-to-limiting-nutrient ratios than similar ecosystems dominated by specialists.*

*-Figure 2: caption needs a more detailed explanation. I guess ESD is derived from the depth integrated biomass? At which depth approximately is carbon export estimated? Are DIC and PO4 concentrations for the surface layers only? Or are they integrated over a specific depth range or the entire water column?*

We now note at line 64 that EcoGEnIE only resolves ecology within the 80 m surface layer, and that carbon export is from this layer to depth.

*For reasons of computational efficiency, EcoGEnIE only represents plankton communities in the 80m surface layer, with no growth below this. Organic matter is produced through mortality and unassimilated grazing, with the relative fractions assigned to dissolved and particulate organic matter (DOM and POM) determined by organism size, with smaller organisms producing a larger fraction of DOM. POM is not explicitly represented, instead being instantly exported from the surface layer and remineralised at depth according to a depth-dependent exponential profile.*

-Figure 2 caption "Values greater than one (red) indicate regions where mixotrophy increased the variable in question, whereas values less than one (blue) indicate a decline." This seems to only apply to panels m-p, the others are centered on 0. Also, I guess the colorbar in panels m-p is in log scale(?) This makes it a bit tricky to read the magnitude of change other than 10 times smaller/larger, because the ticks are not evenly distributed and are not shown in the colorbar. Consider adding some ticks or more labels (in-between 10, 0 and 0.1).

We have added more ticks to the relevant color scale and have added the following text to the legend:

*All rows show Mixotrophy-Two-Guild (on a linear colour scale), except C:P uptake, which shows Mixotrophy÷Two-Guild (on a logarithmic colour scale).*

[Figure]

-Figure 2, it would be good to have a figure (perhaps in an appendix) showing the base-line values of each panel (e.g. the actual ESD values across latitudes for the mixotrophic run and for the separate guild run etc), so that the reader knows whether the absolute differences shown in the figure are large or not. Otherwise, showing the relative differences (e.g. in percent change) could work too.

We have added a new Figure A.2 to an Appendix. This shows the baseline values for the Two-Guild configuration. This provides sufficient information to appreciate the relative magnitude of the absolute changes.

-L131 "Contrary to Fig. 1, in which large and stable differences between the Two-Guild and Mixotrophic are established within the first year of the simulations, noticeable difference between the two configurations only begin to emerge on multi-decadal timescales (green colours)." This sentence is somewhat confusing, isn't the difference simply because these two figures show two different things? i.e. figure 1 shows surface

*only while figure 3 depth resolved differences? I guess the sentence is just confusing and needs to be rephrased (i.e. instead of saying that the difference is between the figures, mention that the difference is between surface and deep ocean).*

We have rewritten the second part of this sentence, switching from

> *"noticeable difference between the two configurations only begin to emerge on multi-decadal timescales."*

to

> *"noticeable differences only begin to emerge at depth on multi-decadal timescales."*

This emphasizes the reviewers point that the difference is between surface and deep ocean

*-L136 "The response within the Atlantic Ocean is more complex. Over the first 500 years, mixotrophy leads to increased DIC storage at all depths below the thermocline, but after about 1000 years, there is a decline in DIC storage between about 1,500 and 3,500 m." but these differences are barely noticeable. Is it really that relevant? Or am I missing something? I guess the interesting point here is that the Atlantic Ocean barely sees any difference while the other two oceans show larger differences.*

These differences are subtle, but we flag them here because they set up the point that the mid-depths in the Atlantic are influenced over longer timescales by the influence of preformed DIC in the North Atlantic Deep Water.

> *The differential response in the Atlantic Ocean occurs at depths associated with the North Atlantic Deep Water (NADW) (Toggweiler and Key, 2001), which suggests some feedback with the ocean circulation. This is explored in Figures 4 and 5, which show the temporal evolution of differences in zonally-averaged chemical profiles for DIC and $PO_4$.*

*-Figure 6. A bit subtle, but it seems to me that DIC total in panel c is "redder" than DIC remineralised from panel a, is this correct? If DIC preformed decreases, shouldn't DIC total be lower than DIC remin?*

This was a subtle difference, but a very important one. We thank the reviewer for their attention to detail. Reviewing our code, we established that we had neglected the contribution of the carbonate counter pump to the remineralised component of DIC. We have corrected this in Figure 6, and in the other values provided in the Abstract and Table 2 for remineralised DIC. Figure 6 now looks like this:

[Figure]

*-Table 2 caption. Mention that these differences correspond to the ones found at the end of the 10K years. Also, for consistency with the other figures, switch "exported" DIC with "remineralised" DIC. A similar table showing the differences in the exported carbon and PO4 at a given depth would also have been interesting.*

We have updated the Table legend, which mow reads as follows:

*Comparison of remineralised, preformed and total DIC (Pg) across the different simulations at year 10,000. The Δ columns are the differences between the four mixotrophic simulations and the Two-Guild simulation.*

**RC2**

*-How does export flux work? Is there a relationship between organism size and sinking rate? Is there sinking of detrital particles? Is there any particle aggregation?*

We have now clarified this point in the Model Description section:

> *For reasons of computational efficiency, EcoGEnIE only represents plankton communities in the 80m surface layer, with no growth below this. Organic matter is produced through mortality and unassimilated grazing, with the relative fractions assigned to dissolved and particulate organic matter (DOM and POM) determined by organism size, with smaller organisms producing a larger fraction of DOM. POM is not explicitly represented, instead being instantly exported from the surface layer and remineralised at depth according to a depth-dependent exponential profile. DOM does not sink.*

*-How does cellular stoichiometry work? Is stoichiometry fixed or flexible?*

We have now clarified this point in the Model Description section:

> *Flexible cellular stoichiometry is included in the model as each population is represented in terms of its C, P and Fe biomass. The proportion of these elements may change within the cell as they are acquired through photosynthesis, the uptake of inorganic nutrients, and/or the ingestion of prey. Ward & Follows (2016) showed that mixotrophs can achieve the same net growth rate with lower internal reserves of P and Fe, supported by their additional flexibility to gain carbon from both photosynthesis and grazing. Mixotrophic ecosystems are therefore able to sustain higher carbon-to-limiting-nutrient ratios than similar ecosystems dominated by specialists.*

*-How many potentially limiting nutrients are tracked and included when modeling growth rates (e.g., N, P, Fe)? The authors focus on gradients of DIC and phosphate, and never discuss nitrate. Is there a reason for this? Do the phosphate and nitrate patterns tell the same story? Nitrogen limits phytoplankton growth more commonly than phosphorus, so it seems important to have some discussion of nitrate.*

EcoGEnIE does not include an explicit representation of the marine nitrogen cycle. This is now noted in the Model Description section:

> *Note that phosphate and iron are the only explicitly resolved nutrients in EcoGEnIE. While the exclusion of nitrogen is not realistic, phosphate provides a reasonable representation of global macronutrient limitation.*

*-The authors interpret the effects of mixotrophy on phosphate in high latitude regions in terms of iron limitation. Specifically, their hypothesis is that mixotrophy alleviates iron limitation, leading to greater phosphate drawdown. This is an interesting idea that would benefit from some more exploration. Which regions of the ocean are iron-limited in this model? And is there more evidence from the model output, such as the growth limitation terms, that can shed light on whether mixotrophy alters iron limitation?*

We have removed the reference to Iron limitation at this point. While we agree that it is an interesting point, we have decided to keep the focus of the paper on carbon and a single limiting nutrient (phosphorus)

*.-The Sensitivity to Tradeoffs section belongs in the results section, not the discussion.*

We have moved this section into the Results.